# LoopFormer: Elastic-Depth Looped Transformers for Latent Reasoning via Shortcut Modulation

**Ahmadreza Jeddi**[1,2,3]    **Marco Ciccone**[1,2]    **Babak Taati**[1,2,3]
[1]University of Toronto    [2]Vector Institute    [3]University Health Network
Corresponding Author: `ajeddi@cs.toronto.edu`

## Abstract

Looped Transformers have emerged as an efficient and powerful class of models for reasoning in the language domain. Recent studies show that these models achieve strong performance on algorithmic and reasoning tasks, suggesting that looped architectures possess an inductive bias toward latent reasoning. However, prior approaches fix the number of loop iterations during training and inference, leaving open the question of whether these models can flexibly adapt their computational depth under variable compute budgets. We introduce Loop-Former, a looped Transformer trained on variable-length trajectories to enable budget-conditioned reasoning. Our core contribution is a shortcut-consistency training scheme that aligns trajectories of different lengths, ensuring that shorter loops yield informative representations while longer loops continue to refine them. LoopFormer conditions each loop on the current time and step size, enabling representations to evolve consistently across trajectories of varying length rather than drifting or stagnating. Empirically, LoopFormer demonstrates robust performance on language modeling and reasoning benchmarks even under aggressive compute constraints, while scaling gracefully with additional budget. These results show that looped Transformers are inherently suited for adaptive language modeling, opening a path toward controllable and budget-aware large language models. Project Page: https://loopformer.github.io

## 1 Introduction

Transformers with parameter sharing, often called *looped* or *recurrent* Transformers, have emerged as an efficient and capable alternative to deep non–shared stacks across vision and natural language Dehghani et al. (2018); Lan et al. (2019); Jaegle et al. (2021); Dutta et al. (2021); Geiping et al. (2025). In particular, looped Transformers in the language modeling setting have shown promising performance on a variety of algorithmic and reasoning tasks Geiping et al. (2025); Saunshi et al. (2024; 2025); Gatmiry et al. (2024). These models appear to possess an inductive bias toward reasoning: a property sometimes referred to as *latent reasoning*, where they internalize reasoning skills akin to explicit chain-of-thought prompting in large language models. Moreover, studies indicate that such abilities scale gracefully with effective computational depth and number of loops, yielding improved results on reasoning benchmarks Xu & Sato (2024); Saunshi et al. (2025); Geiping et al. (2025). However, existing approaches almost always train and evaluate with a fixed number of unrolls. This raises a fundamental question: do looped Transformers truly exploit their computational depth flexibly, and can they be trained to operate effectively under variable compute budgets?

Despite their promise, current looped models remain tied to a single trajectory length. Once trained, their representations collapse when evaluated at shorter or longer depths, since those settings are off-distribution Bae et al. (2024); Fan et al. (2024). In practice, this means looped models spend the same budget as non-looped iso-FLOP baselines, forfeiting one of their key motivations: flexible compute. We instead consider *budget-conditioned* language modeling: at inference, a user specifies a compute budget $M$, and the model should produce high-quality representations without retraining. While early exiting, routing, and layer dropping have made non-looped Transformers more dynamic

Schuster et al. (2022); Fan et al. (2019); Elhoushi et al. (2024); Shazeer et al. (2017); Raposo et al. (2024), little has been explored for looped models. Naively transplanting these techniques into *looped* architectures is fragile Bae et al. (2025); Geiping et al. (2025): repeated passes of the shared block often converge to similar, *stagnant* states. We aim for *elastic depth*: a single looped model that performs well across user-chosen budgets without retraining or late-step degeneracy. The core challenge is to train looped models whose internal trajectories remain *stable across depths*, so that shorter routes do not degrade and longer routes continue to refine rather than collapse.

We present **LoopFormer**, a shortcut-modulated looped language model that supports elastic depth, maintaining strong performance across a range of inference budgets. Inspired by diffusion models Frans et al. (2024); Lu & Song (2024) and neural ODEs Chen et al. (2018), we cast iterative representation refinement as a trajectory in representation space: token states evolve from an initial $h_0$ toward a target $h_1$ over a normalized unit-time horizon. Our key insight is to explicitly condition each loop step on the current time $t$ and a step size $\Delta t$ (a "jump"), allowing coarser trajectories to approximate fine-grained ones with fewer steps. Training employs a *shortcut-consistency* objective that aligns the final representations of shorter routes with those of the full route via a stop-gradient target, effectively performing self-distillation within the loop. At inference, this conditioning yields *elastic depth* without retraining: the user selects a budget $M \leq L$ (maximum loops) and a step schedule, and performance scales smoothly with compute, rather than collapsing at shorter depths.

Departing from dynamic compute through routing or token halting Dehghani et al. (2018); Bae et al. (2025), we instead introduce the notion of *loop trajectories*, which enable effective and efficient language modeling, thereby giving the model the ability to internalize and refine reasoning and performance as compute increases. Empirically, LoopFormer not only preserves the language modeling abilities reported in prior work, but also outperforms baseline looped models on language tasks and closes much of the gap with iso-FLOP non-looped models. This creates an opportunity to study loop trajectories as a new lens on reasoning in looped models. Moreover, unlike early-exit mechanisms that often collapse to degenerate states, LoopFormer maintains stable refinement: both perplexity and reasoning scale gracefully with the compute budget.

The key contributions of this paper are summarized as follows:

– We formulate a class of shortcut-modulated looped Transformers for language modeling that conditions each pass on internal time and step sizes.

– We introduce a shortcut-consistency training protocol over families of step schedules that enables compute-budgeted inference (elastic depth) without retraining.

– We analyze representation dynamics across loop iterations using multiple geometric and information-theoretic metrics (anisotropy, curvature, entropy, CKA), finding that adaptive early-exit looped models exhibit representational collapse, while our shortcut-modulated models maintain evolving, non-degenerate states.

– We demonstrate consistent performance-per-compute gains in both perplexity and zero-shot reasoning across diverse language benchmarks; ablations isolate the roles of time/step conditioning and offer practical guidance for choosing trajectories under fixed budgets.

## 2 RELATED WORK

**Recursive / Looped Transformers.** Parameter sharing provides an orthogonal route to efficiency and effective depth. The Universal Transformer demonstrated that repeatedly applying a single block can match the representational capacity of deep non–shared stacks, while also introducing adaptive computation time Dehghani et al. (2018). ALBERT further showed that extensive cross-layer weight-tying yields substantial parameter savings during pretraining, without compromising downstream performance Lan et al. (2019). Deep Equilibrium Models (DEQ) extend this paradigm by defining an implicit infinitely deep, weight–tied transformation solved via fixed-point iteration and implicit differentiation Bai et al. (2019), with related equilibrium and recurrent architectures also explored in vision and multimodal contexts Jaegle et al. (2021); Yang et al. (2022). More recent work has studied looping as programmable computation Giannou et al. (2023), iterative data-fitting or optimization solvers Yang et al. (2023), mechanisms for algorithmic length generalization Fan et al. (2024), and even the Turing-completeness of looped decoders for certain graph algorithms De Luca & Fountoulakis (2024). Most relevant to our work, *Time-Modulated Looped Transformers* (TMLT)

Xu & Sato (2024) analyze the expressive power of looped Transformers in language modeling, showing that conditioning on timestep (loop index) improves scaling and perplexity. Our approach builds on this insight but extends it by conditioning on both normalized time and step size, and by training across families of trajectories rather than a single fixed schedule.

**Dynamic compute.** Dynamic compute allocation reduces inference cost by skipping or reallocating computation where it is not needed Bengio et al. (2015); Huang et al. (2016); Panda et al. (2016). Early exiting halts processing for "easy" inputs at intermediate layers, allowing deeper computation only for harder cases Elbayad et al. (2019); Schuster et al. (2022); Elhoushi et al. (2024). Other approaches include layer dropping and pruning strategies for BERT-style models Fan et al. (2019); Hou et al. (2020). Mixture-of-Experts (MoE) increases capacity through sparse expert routing Shazeer et al. (2017); Fedus et al. (2022), while *Mixture-of-Depths* (MoD) Raposo et al. (2024) reframes adaptivity as token-wise routing across layers, enabling fine-grained dynamic allocation of depth. Despite these advances, the adaptation of dynamic compute techniques to exploit the looping capability of recursive models has received little attention. A concurrent line of work, *Mixture of Recursions* Bae et al. (2025), extends routing to recursive stacks by varying the number of shared-block applications per token. In contrast, our approach does not route or halt computation at the token level; instead, we train looped models to be *budget-conditioned*, ensuring that the same parameters operate robustly across user-specified loop budgets.

**Latent reasoning.** An increasing body of work investigates reasoning carried out within hidden states rather than through explicit chain-of-thought prompting Goyal et al. (2023); Cheng & Van Durme (2024); Pfau et al. (2024); Kaissis et al. (2026); Chen et al. (2026); Zhu et al. (2025); Jolicoeur-Martineau (2025); Wang et al. (2025); Zhang et al. (2025); Knupp et al. (2026); Liang & Pan (2026). Several approaches employ looped models to simulate multi-step reasoning or to approximate chain-of-thought dynamics directly Hao et al. (2024); Saunshi et al. (2025); Wang et al. (2025). In parallel, theoretical and empirical studies connect network depth and algorithmic generalization to reasoning ability Merrill & Sabharwal (2023); Chen & Zou (2024); Ye et al. (2024); Xu & Sato (2025). More recently, analyses suggest that looped Transformers possess an inductive bias for reasoning that strengthens with increasing effective computational depth Saunshi et al. (2024; 2025). Collectively, this line of work frames such abilities as *latent reasoning*, wherein models perform iterative computations within their hidden-state space without explicit verbalization. Our work leverages this inductive bias of looped models but emphasizes *trajectory robustness*: ensuring that hidden-state computation remains informative even under shorter budgets, while continuing to refine effectively at greater depths.

**Time / shortcut modulation and conditioning.** Conditioning networks on continuous *time* or related control variables has proven highly effective in generative diffusion modeling Ho et al. (2020); Rombach et al. (2022); Lipman et al. (2022). Diffusion Transformers (DiT) Peebles & Xie (2023) incorporate timestep-conditioned adaptive normalization, while consistency-based training aligns solutions across discretizations Song et al. (2023). Recent shortcut and one-step diffusion approaches further distill long trajectories into a few steps by enforcing consistency between coarse and fine solvers Frans et al. (2024); Lu & Song (2024). These ideas are increasingly migrating into language modeling: for example, *Time-Modulated Looped Transformers* (TMLT) Xu & Sato (2024) adapt DiT-style timestep conditioning to recursive language models, demonstrating improved scaling and perplexity. LoopFormer extends this trend by conditioning each loop not only on normalized time $t$ but also on the step size $\Delta t$, and by training across families of sampled trajectories with a shortcut-consistency objective. This design enables robust performance under arbitrary user-specified compute budgets, without relying on per-token halting or routing mechanisms.

## 3 SHORTCUT-MODULATED LOOPED MODELS FOR LATENT REASONING

### 3.1 NOTATION AND PROBLEM STATEMENT

We denote by $X = (x_1, \ldots, x_T)$ a sequence of $T$ tokens drawn from a vocabulary $\mathcal{V}$. The token embeddings are obtained via $E_{\text{tok}}(X) \in \mathbb{R}^{T \times d}$. Without loss of generality, positional embeddings can be added in a one-shot manner or applied through alternatives such as RoPE; in this work, we

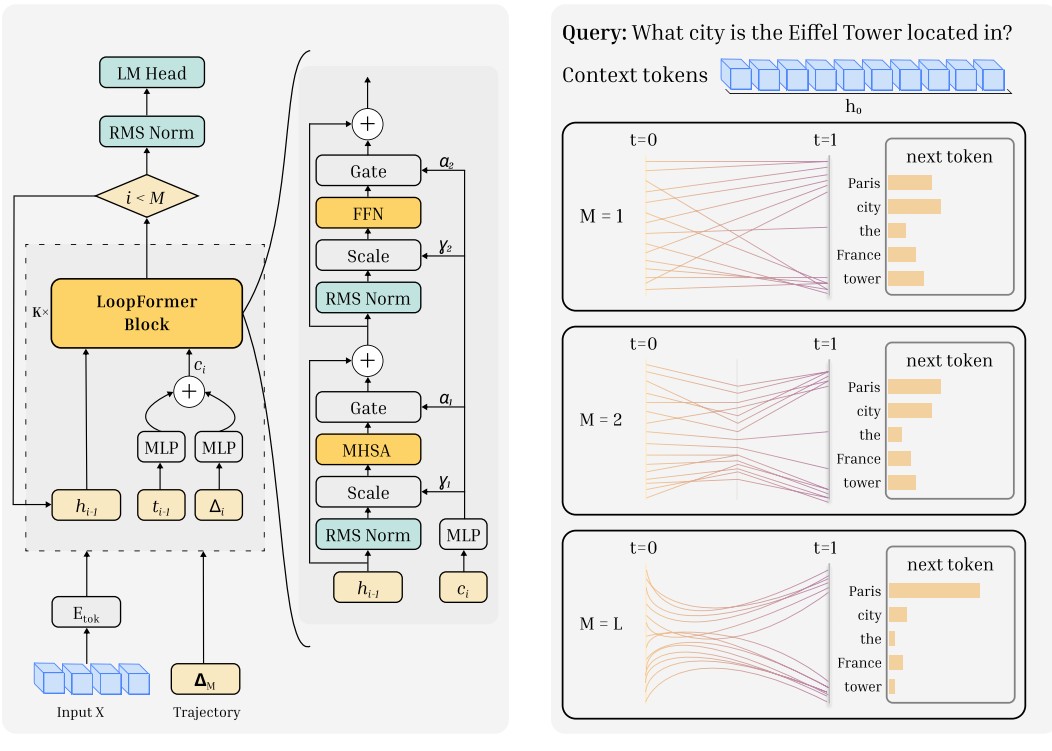

(a) LoopFormer architecture  (b) Budget-conditioned trajectories

Figure 1: **(a)** LoopFormer conditions each loop on normalized time $t$ and step size $\Delta t$, modulating RMSNorm scales and gating the MHSA/FFN residuals. **(b)** During inference, shorter-budget trajectories ($M < L$) approximate the full $L$-step route; more budget yields progressively refined next-token distributions while preserving utility at low budget.

adopt the former for simplicity. The initial hidden states are therefore

$$h^{(0)} = E_{\text{tok}}(X) + E_{\text{pos}}[1{:}T] \in \mathbb{R}^{T \times d}.$$

A range of looping mechanisms for looped Transformers has been studied in prior work Takase & Kiyono (2021); Saunshi et al. (2025); Bae et al. (2025; 2024), including cycle, middle-cycle, and relaxed-cycle variants. Since our focus is on trajectories and their representation dynamics, we adopt the simplest *cycle* design, where a stack of $k$ Transformer blocks, denoted by $\Phi_k(\cdot)$, is recursively applied on the hidden state. Following Saunshi et al. (2025), we write $(k \otimes L)$ for a looped model with $k$ blocks repeated $L$ times (approximate cost $kL$ FLOPs), and $(kL \otimes 1)$ for a non-looped Transformer of comparable depth.

LoopFormer applies $\Phi_k(\cdot)$ for $M$ iterations ($1 \leq M \leq L$), conditioning each loop $i$ on the pair $(t_{i-1}, \Delta_i)$, where $0 = t_0 < t_1 < \cdots < t_M = 1$ are cumulative normalized timesteps, and $\Delta_i = t_i - t_{i-1}$ is the step size of the $i$-th iteration. We refer to $\boldsymbol{\Delta_M} = (\Delta_1, \ldots, \Delta_M)$ as a *trajectory*, constrained by $\sum_{i=1}^{M} \Delta_i = 1$.

The *maximum trajectory* corresponds to $L$ uniform steps with $\Delta_i = \frac{1}{L}$ for all $i$. At inference, a user specifies a compute budget $M$ and provides a step schedule $\boldsymbol{\Delta_M}$ such that $\sum_{i=1}^{M} \Delta_i = 1$.

### 3.2 ARCHITECTURE

We introduce **LoopFormer**, a looped language modeling architecture designed to remain faithful to the standard Transformer while incorporating trajectory-based conditioning. Figure 1a provides an overview. In this section we describe the forward pass and the core components of the model.

LoopFormer is a looped decoder-only Transformer in which a single shared stack is applied iteratively; the key novelty is *how each loop is conditioned*. At iteration $i$, the model conditions on the pair $(t_{i-1}, \Delta_i)$, where $t_{i-1} \in [0, 1]$ is the cumulative normalized time and $\Delta_i \in (0, 1]$ is the step size. Both scalars are encoded with sine–cosine frequency embeddings and projected by small MLPs to obtain $e_t$ and $e_\Delta$, which are summed to form $e_i = e_t + e_\Delta$. This signal modulates the *LoopFormer Block*: an MLP maps $e_i$ to scaling $(\gamma_1, \gamma_2)$ for the two RMSNorm layers and to gating $(\alpha_1, \alpha_2)$ applied immediately before the residual connections of MHSA and FFN. As a result, each loop is explicitly aware of its location and granularity in the trajectory, enabling consistent behavior across coarse- and fine-grained schedules.

**Relation to prior work.** For the design of our architecture, we follow the overall approach of looped models such as ALBERT and recent looped decoders Lan et al. (2019); Saunshi et al. (2025), and focus our novelty on *trajectory conditioning*. Our modulation mechanism is inspired by Diffusion Transformers (DiT) Peebles & Xie (2023) and shortcut/one-step diffusion Frans et al. (2024), which condition blocks on a timestep via adaptive normalization (adaLN), regressing scale/shift and gates from a time embedding. In the looped language model setting, TMLT Xu & Sato (2024) adopts this DiT-style idea by using the loop index as time (with RMSNorm). LoopFormer extends this conditioning by using both normalized time $t$ and step size $\Delta t$, and by training over families of trajectories with a shortcut-consistency objective, yielding trajectory-consistent representations suitable for budget-conditioned inference.

### 3.3 Training & Inference with LoopFormer

A key goal of LoopFormer is to train models that perform well even with fewer than $L$ loops, thereby enabling *elastic depth* at inference time. Unlike prior adaptive-compute approaches that rely on per-token halting or early exits, our framework uses a user-defined compute budget, similar in spirit to diffusion models. Empirically, we find that naive early exiting in looped architectures leads to *stagnant* representations in later iterations, under-utilizing depth. To address this, LoopFormer leverages shortcut conditioning together with a consistency loss that encourages trajectories of different lengths to converge toward the full trajectory of length $L$. The overall objective is:

$$\mathcal{L} \;=\; \mathcal{L}_{\mathrm{L}} \;+\; \lambda_1\, \mathcal{L}_{\mathrm{S}} \;+\; \lambda_2\, \mathcal{L}_{\mathrm{cons}},$$

where $\mathcal{L}_{\mathrm{L}}$ and $\mathcal{L}_{\mathrm{S}}$ denote next-token prediction losses for the longest and sampled shortcut trajectories, respectively, and $\mathcal{L}_{\mathrm{cons}}$ is a stop-gradient consistency loss aligning per-token logits of shorter trajectories to those of the longest trajectory. We set $\lambda_1 = \lambda_2 = 0.1$ in all experiments. Algorithm 1 formalizes the training procedure.

**Shortcut trajectory sampling.** Given a LoopFormer $(k \otimes L)$ and the maximum trajectory $\mathbf{\Delta_L}$, during training, we additionally sample a shortcut trajectory $\mathbf{\Delta_S}$ with budget $1 \leq S < L$. For simplicity, at each batch we first sample a shortcut length $S \sim \mathcal{U}\{1, \ldots, L-1\}$, then uniformly draw the discrete step schedule $\mathbf{\Delta_S}$ over $[0, 1]$ such that $\sum_{i=1}^{S} \Delta_i = 1$. This ensures exposure to both long and short trajectories during training.

**Inference with elastic depth.** At inference time, LoopFormer can be deployed flexibly at any budget $M \leq L$. A user specifies $M$ and $\mathbf{\Delta_M}$, and the model produces outputs that scale smoothly with compute, without retraining. Figure 1b provides a conceptual illustration of budget-conditioned refinement: for any chosen budget, shorter routes are trained to approximate the $t{=}1$ endpoint, with quality improving as steps increase. Algorithm 2 outlines the inference procedure.

## 4 Experiments

We evaluate latent reasoning, scalability, efficiency, and representation dynamics of our shortcut-modulated looped language models. Following Tay et al. (2022); Saunshi et al. (2025), we compare a 24-layer, $\sim$1B-parameter non-looped Transformer with FLOP-matched looped variants. All models use a GPT-style decoder Radford et al. (2019) with NanoGPT configurations. Training is performed on a deduplicated subset of The Pile Gao et al. (2020) for 25B tokens in accordance with Chinchilla scaling Hoffmann et al. (2022). See Appendix A for details. Unless otherwise specified, all reported results use uniform step sizes at inference: for a compute budget $M$, $\Delta_i = 1/M$ for all $i$.

---

**Algorithm 1** LoopFormer Training

> **while** not converged **do**
>   Sample batch $(X, Y)$
>   Construct max trajectory $\mathbf{\Delta_L}$
>   Sample shortcut length $S \sim \mathcal{U}\{1, \dots, L-1\}$
>   Sample trajectory $\mathbf{\Delta_S}$ of length $S$
>   $h^{(L)} \leftarrow \Phi_k(h^{(0)}; \mathbf{\Delta_L})$ ; $h^{(S)} \leftarrow \Phi_k(h^{(0)}; \mathbf{\Delta_S})$
>   $\mathcal{L}_L \leftarrow \text{CE}(\text{LMHead}(h^{(L)}), Y)$
>   $\mathcal{L}_S \leftarrow \text{CE}(\text{LMHead}(h^{(S)}), Y)$
>   $\mathcal{L}_{\text{cons}} \leftarrow \| \text{stopgrad}(h^{(L)}) - h^{(S)} \|^2$
>   Update $\theta$ using $\nabla_\theta(\mathcal{L}_L + \lambda_1 \mathcal{L}_S + \lambda_2 \mathcal{L}_{\text{cons}})$

---

**Algorithm 2** LoopFormer Inference

> Given input $X$
> Choose budget $M \leq L$
> Sample schedule $\mathbf{\Delta_M}$
> Initialize $h^{(0)} \leftarrow E_{\text{tok}}(X) + E_{\text{pos}}$
> **for** $i = 1$ to $M$ **do**
>   $h^{(i)} \leftarrow \Phi_k(h^{(i-1)}; t_{i-1}, \Delta_i)$
> **return** $\text{LMHead}(h^{(M)}[:, -1])$

---

Table 1: Perplexity and zero-shot reasoning for $(3 \otimes 8)$ looped models under three inference budgets $(24\times, 12\times, 6\times)$. At $24\times$ we also report fixed-depth baselines; at $12\times/6\times$ we compare against Base $(12 \otimes 1)$ and $(6 \otimes 1)$. While depth-elastic, LoopFormer narrows the perplexity gap to Base and is competitive on reasoning, outperforming other looped variants, especially at higher budgets.

| | Params / FLOPs | Perplexity ↓ | | | Language Tasks (Accuracy) ↑ | | | | | | | | | | |
|---|---|---|---|---|---|---|---|---|---|---|---|---|---|---|---|
| | | Pile | FineWeb-Edu | OpenWebText | COPA | HS | LB | OBQA | PIQA | Race | SciQ | ARC | SIQA | WG | Avg Acc |
| **Budget: 24x** | | | | | | | | | | | | | | | |
| Base $(24 \otimes 1)$ | 24x / 24x | **9.49** | **20.7** | **20.08** | 61 | **35.04** | **41.96** | 27.6 | **66** | 29 | **70.1** | **33.43** | _38.18_ | 50.4 | 45.27 |
| Base-Loop $(3 \otimes 8)$ | 3x / 24x | 10.91 | 24.53 | 24.53 | 61 | 30.46 | 34.68 | 27 | 63.22 | 28.8 | 63.7 | 31.71 | **38.43** | 49.8 | 42.88 |
| TMLT $(3 \otimes 8)$ | 3x / 24x | 10.38 | _22.87_ | 21.99 | 65 | _32.34_ | _39.06_ | _27.8_ | 63.11 | _29.67_ | _69.8_ | 31.67 | 36.95 | 51.54 | 44.69 |
| Naive-Loop-EE $(3 \otimes 8)$ | 3x / 24x | 11.6 | 26.55 | 25.64 | **66** | 29.68 | 31.52 | 27 | 62.24 | 28.33 | 65.5 | 31.64 | 36.64 | 50.59 | 42.91 |
| Base-Loop-EE-Cons $(3 \otimes 8)$ | 3x / 24x | 11.56 | 25.33 | 24.41 | **66** | 30.54 | 32.19 | 26.8 | 62.13 | 28.23 | 64.9 | 31.45 | 37.4 | **52.88** | 43.25 |
| TMLT-EE $(3 \otimes 8)$ | 3x / 24x | 10.7 | 24.07 | 23.17 | 65 | 31.03 | 35.14 | **28.4** | 63.11 | 28.8 | 68.8 | 31.74 | 37.41 | 50.83 | 44.03 |
| LoopFormer - Ours $(3 \otimes 8)$ | 3x / 24x | _10.28_ | _22.87_ | _21.98_ | **66** | 32.3 | 38.27 | 26.8 | _63.33_ | 30.81 | 68 | _32.71_ | 37.97 | _51.94_ | 44.81 |
| **Budget: 12x** | | | | | | | | | | | | | | | |
| Base $(12 \times 1)$ | 12x / 12x | **9.98** | **22.24** | **21.41** | _67_ | **32.72** | **37.78** | 26.2 | **64.69** | **29.86** | **69.1** | **32.29** | **38.38** | _51.3_ | 44.93 |
| Naive-Loop-EE $(3 \otimes 4)$ | 3x / 12x | 11.66 | 26.74 | 25.81 | 65 | 29.35 | 31.28 | 26 | 61.75 | 28.32 | 64.4 | 31.15 | 36.8 | 51.85 | 42.59 |
| Base-Loop-EE-Cons $(3 \otimes 4)$ | 3x / 12x | 12.0 | 27.72 | 26.68 | 63 | 29.72 | 25.6 | **27** | 61.26 | 28.04 | 58.4 | 30.06 | 36.23 | 51.7 | 41.1 |
| TMLT-EE $(3 \otimes 4)$ | 3x / 12x | 12.18 | 28.28 | 27.11 | 60 | 29.88 | 27.73 | _26.4_ | 62.02 | _28.8_ | 61.4 | 30.59 | 36.69 | 51.54 | 41.5 |
| LoopFormer - Ours $(3 \otimes 4)$ | 3x / 12x | _11.12_ | _25.02_ | _24.21_ | **68** | _31_ | _32.35_ | 25.4 | _63.06_ | 28.23 | _66.3_ | _31.78_ | _37.77_ | **53.43** | 43.73 |
| **Budget: 6x** | | | | | | | | | | | | | | | |
| Base $(6 \times 1)$ | 6x / 6x | **11.13** | **25.28** | **24.28** | **64** | **30.45** | **33.45** | _25.6_ | **62.51** | **28.04** | **67.5** | **31.07** | **36.18** | 48.54 | 42.73 |
| Naive-Loop-EE $(3 \otimes 2)$ | 3x / 6x | _12.61_ | _29.36_ | _28.38_ | _63_ | 28.64 | _27.28_ | 24.6 | _61.48_ | _26.41_ | _62.3_ | 29.91 | _35.41_ | _50.67_ | 40.97 |
| Base-Loop-EE-Cons $(3 \otimes 2)$ | 3x / 6x | 15.07 | 35.95 | 34.88 | 59 | 28.28 | 18.49 | 25.6 | 59.52 | 25.16 | 53.8 | 28.94 | 34.65 | **52.72** | 38.62 |
| TMLT-EE $(3 \otimes 2)$ | 3x / 6x | 15.79 | 37.83 | 37.84 | 57 | 28.18 | 17.09 | 25.08 | 58.68 | _26.41_ | 50.01 | 28.26 | 34.9 | 50.27 | 37.59 |
| LoopFormer - Ours $(3 \otimes 2)$ | 3x / 6x | 14.3 | 33.45 | 32.46 | _63_ | _28.69_ | 26.14 | **26.6** | 60.1 | 25.74 | 58.6 | _29.29_ | 35.26 | 50.2 | 40.36 |

## 4.1 LATENT REASONING AND PERPLEXITY

We train looped models under different parameter and compute budgets. Using the notation $(k \otimes L)$ for a $k$-block $\Phi_k$ unrolled $L$ times, we consider $k \in \{1, 2, 3\}$ and train with maximum loops $L \in \{8, 12, 24\}$. Fixed-loop models are trained and evaluated with the same $L$, whereas depth-elastic models support inference at any $M \leq L$. In this section we use $kL$ as a proxy for FLOPs when comparing baselines, ignoring embedding/unembedding costs; Table 1 summarizes the results for $k = 3$.

**Evaluation metrics and benchmarks.** Following Saunshi et al. (2025); Geiping et al. (2025); Bae et al. (2025), we report perplexity and downstream zero-shot accuracy. Perplexity is measured on FineWeb-Edu Penedo et al. (2024), OpenWebText Gokaslan & Cohen (2019), and The Pile. For

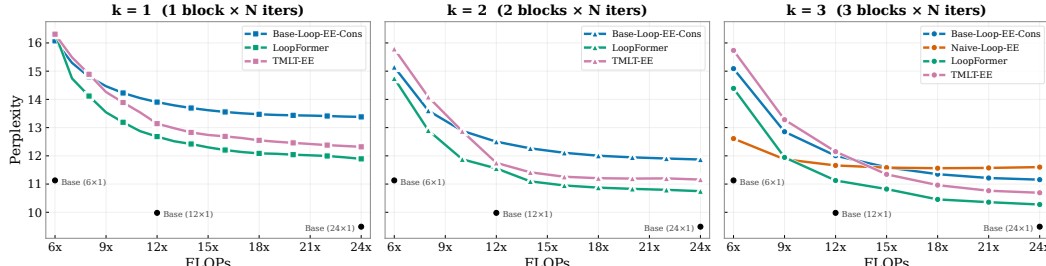

Figure 2: Scaling with layers and loops. Perplexity on The Pile across $(k, L)$ and inference budgets $M \leq L$; larger $k$ lowers perplexity at fixed $M$, and additional loops further reduce it.

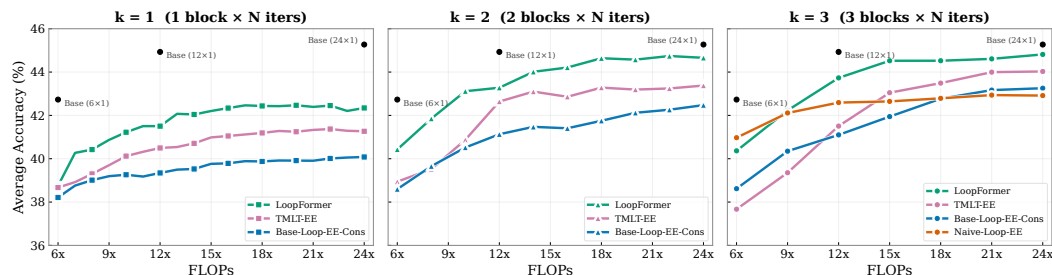

Figure 3: Scaling with layers and loops. Average zero-shot reasoning accuracy on 10 tasks for the same settings; more loops improve reasoning at fixed $k$. Across budgets, LoopFormer scales smoothly without collapse and consistently outperforms other looped baselines.

latent reasoning, we report zero-shot accuracy on ten established benchmarks spanning a range of reasoning difficulty: COPA Roemmele et al. (2011), HellaSwag (HS) Zellers et al. (2019), LAMBADA (LB) Paperno et al. (2016), OpenBookQA (OBQA) Mihaylov et al. (2018), PIQA Bisk et al. (2020), RACE Lai et al. (2017), Social IQA (SIQA) Sap et al. (2019), ARC Clark et al. (2018), SciQ Welbl et al. (2017), and WinoGrande (WG) Sakaguchi et al. (2021). Where available, we use normalized accuracy; for ARC we report the average over Easy and Challenge.

**Baselines.** We compare LoopFormer against two groups of baselines.

*Fixed-depth models:* (a) **Base**: a non-looped Transformer; (b) **Base-Loop**: a standard looped model as in Saunshi et al. (2025); (c) **TMLT**: a looped model with timestep conditioning Xu & Sato (2024).

*Depth-elastic models:* (i) **Base-Loop-EE**: naive early exiting applied to the basic looped model;[1] (ii) **Base-Loop-EE-Cons**: (i) augmented with our consistency loss during training; (iii) **TMLT-EE**: early exiting and consistency training applied to TMLT to enable depth elasticity.

**Findings.** Table 1 highlights three trends:

– Consistent with Mohtashami et al. (2023); Saunshi et al. (2025), looped models trail non-looped baselines on perplexity, reflecting the role of parameters in memorization. However, LoopFormer closes much of this gap, surpassing even fixed-depth looped variants.

– In zero-shot reasoning, looped models benefit from iterative refinement and can approach iso-FLOP non-looped baselines; LoopFormer is the most competitive among looped models.

– Under reduced budgets, LoopFormer maintains high utility: at $12\times$ it remains close to Base $(12 \otimes 1)$ on both perplexity and reasoning, indicating that budget-conditioned trajectories preserve informative representations rather than collapsing.

We next examine the effect of the number of layers $k$ and the number of loops $L$, and compare against depth-elastic alternatives under more compute configurations.

---

[1]Akin to Recurrent Depth Geiping et al. (2025) without sandwich normalizations or a randomly initialized recurrent state. We trained both variants; the simpler one performs better.

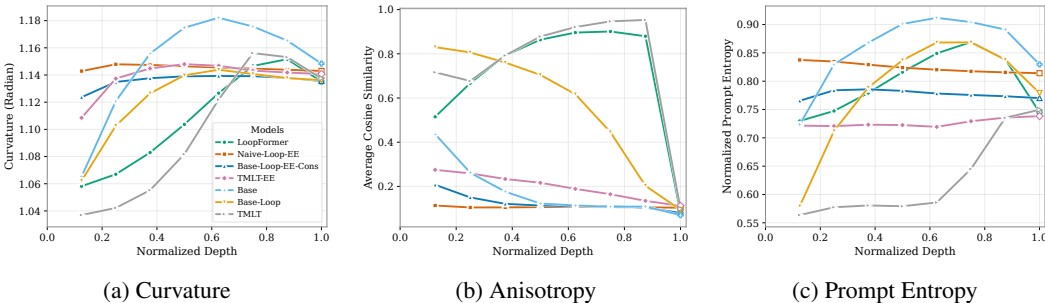

|     |     |     |
| --- | --- | --- |
| (a) Curvature | (b) Anisotropy | (c) Prompt Entropy |

Figure 4: Representation metrics over normalized depth. Panels show **(a)** curvature, **(b)** anisotropy, and **(c)** normalized prompt entropy. Early-exit baselines remain flat, indicating minimal change with additional loop steps, whereas LoopFormer exhibits sustained evolution that rises through mid-depths and tapers near the end, suggesting useful depth-elastic dynamics.

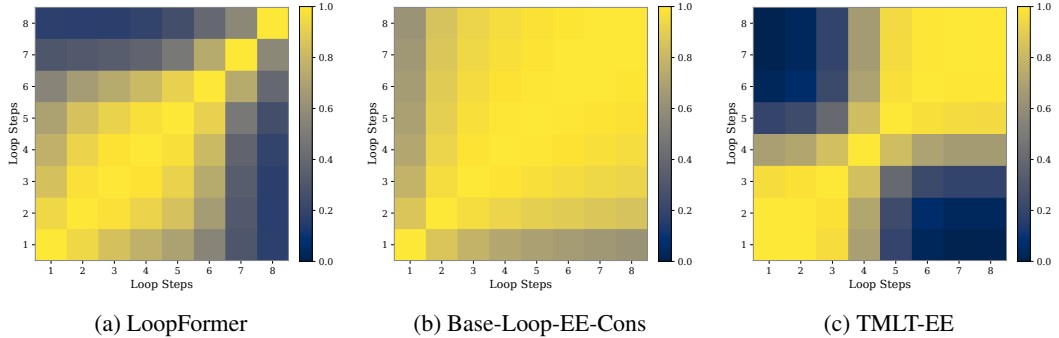

|     |     |     |
| --- | --- | --- |
| (a) LoopFormer | (b) Base-Loop-EE-Cons | (c) TMLT-EE |

Figure 5: CKA similarity across loop steps. Each heatmap reports cross-step CKA within a model family. Depth-elastic baselines show high CKA (indicating little change across loops), suggesting stagnation. LoopFormer exhibits progressive drift, especially toward the later steps.

## 4.2 NUMBER OF LAYERS VS. NUMBER OF LOOPS

We study how perplexity and reasoning change as the number of Transformer layers per block ($k$) varies. We train $k \in \{1, 2, 3\}$ with maximum loops $L \in \{8, 12, 24\}$, then evaluate across multiple inference budgets ($M \leq L$). Figure 2 reports perplexity and Figure 3 reports average zero-shot reasoning accuracy across benchmarks.

As the plots show, both perplexity and reasoning improve with larger $k$ and more iterations. **Loop-Former** preserves these trends under budgets ($M \leq L$): shorter trajectories remain informative and added loops yield smooth gains without collapse. Our consistency-augmented training also improves the scaling of Base-Loop and TMLT in the depth-elastic regime.

## 4.3 ANALYZING REPRESENTATION COLLAPSE IN LOOPED TRANSFORMERS

The role of depth in Transformers has been widely examined through scaling laws and theory Kaplan et al. (2020); Csordás et al. (2025), with a recurring observation that very deep stacks can exhibit *representation degeneration* (or *collapse*), where hidden states change little across layers Ethayarajh (2019); Dong et al. (2021); Godey et al. (2024). Several works attribute this to an inductive bias of self-attention toward uniformity, including rank decay in attention maps with depth Dong et al. (2021). Related studies of *anisotropy* Razzhigaev et al. (2023); Godey et al. (2024) report pronounced effects in language modeling, and these effects could be amplified in looped architectures where a single block is repeatedly applied Dong et al. (2021).

We analyze token dynamics along the computation depth using four complementary metrics: (i) **anisotropy** Godey et al. (2024) within a layer, measured as average pairwise cosine similarity among tokens in the prompt; higher values indicate more aligned (less diverse) directions. (ii) **Curva-**

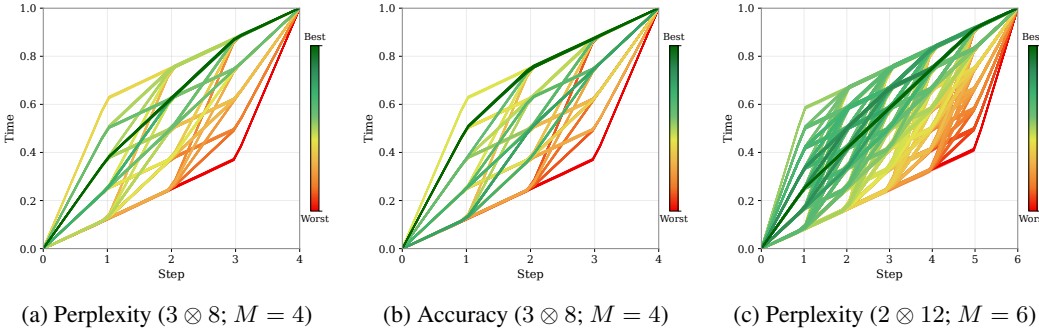

(a) Perplexity $(3 \otimes 8; M = 4)$     (b) Accuracy $(3 \otimes 8; M = 4)$     (c) Perplexity $(2 \otimes 12; M = 6)$

Figure 6: Performance across length-$M$ trajectories. **(a,b)** enumerate all $M=4$ schedules for Loop-Former $(3 \otimes 8)$, reporting perplexity and average zero-shot accuracy; **(c)** shows perplexity for $M=6$ schedules of $(2 \otimes 12)$. Even at fixed budget, trajectory choice matters: spreads are large, and top schedules allocate coarser steps early and finer steps late.

**ture** Hosseini & Fedorenko (2023), a local geometric measure of how rapidly token representations change direction across neighboring positions. (iii) **Prompt entropy** Skean et al. (2025), a matrix-based estimate of how spread-out token embeddings are across feature dimensions; higher entropy suggests greater diversity and lower redundancy. (iv) **CKA similarity** Kornblith et al. (2019) across loop steps, quantifying representational similarity between different iterations of the shared block. While prior work studies these metrics in various architectures and their correlation with downstream performance Garrido et al. (2023); Skean et al. (2025), here we use them primarily as diagnostics to assess whether looped models *use* additional computation (iterative depth) or *stagnate*.

Figure 4 summarizes curvature, anisotropy, and prompt entropy across loop steps; Figure 5 reports cross-step CKA. Early-exit looped baselines show flat trajectories on all three metrics and high CKA, indicating stagnation. In contrast, LoopFormer evolves: early steps have lower curvature, weaker angular alignment, and lower entropy; as steps progress, curvature and entropy rise and alignment increases, then all three taper near the final step as the model prepares for unembedding.

Overall, these patterns suggest that LoopFormer maintains *useful* representational dynamics across loops, with shorter trajectories remaining informative and longer trajectories providing additional refinement, rather than converging prematurely to a static state. We view this as evidence that shortcut conditioning and consistency training help avert collapse in depth-elastic looped models.

### 4.4 How to choose trajectories under a fixed budget

A practical question for our depth-elastic models is: *given a budget $M \leq L$, how should we choose* $\mathbf{\Delta}_M$? Beyond average gains, we ask which *trajectories* work best and which parts of time contribute most. The representation analyses in Figure 4 and Figure 5 suggest a pattern: early steps produce more similar states, activity rises mid/late, then tapers near the end.

To probe this directly, we run a toy yet exhaustive study. For LoopFormer $(3 \otimes 8)$ with budget $M = 4$, we enumerate all trajectories with step sizes summing to 1 (aligned to training granularity) and measure perplexity and average zero-shot accuracy (Figure 6a, Figure 6b). Despite identical compute, performance varies across schedules (spread $\sim$1.4 perplexity and $\sim$1.3 accuracy points). Repeating for $(2 \otimes 12)$ with $M = 6$, the perplexity spread grows to nearly 3 (Figure 6c).

**Findings.** (1) Even under uniform training over budgets and times, some schedules outperform others by wide margins. (2) The best schedules for perplexity and for downstream reasoning are close but not identical; both favor allocating larger steps early and finer steps late.

## 5 Discussion and Future Directions

We introduced LoopFormer, a looped Transformer that conditions each iteration on normalized time $t$ and step size $\Delta t$, trained across trajectories with a shortcut–consistency loss. This framing induces

*loop trajectories* in hidden space: under a fixed budget, shorter routes yield useful intermediate states, and additional steps refine them toward a shared $t{=}1$ endpoint, realizing latent reasoning while supporting budget–conditioned (elastic) inference without retraining. Empirically, Loop-Former delivers strong performance–per–compute on perplexity and consistently improves downstream zero-shot reasoning, while avoiding the representational stagnation seen in naive early-exit baselines. Limitations include global (sequence-level) rather than instance/token-adaptive budgeting, added training overhead from multi-trajectory consistency, and correlational (not causal) representation analyses. Promising directions include instance-conditioned schedule policies and deeper theory/diagnostics of the representation space.

## 6 ACKNOWLEDGEMENTS

We thank Prof. Colin Raffel (University of Toronto) and Prof. Ali Etemad (Queen's University) for valuable feedback throughout this project. We also thank Negin Baghbanzadeh and Dev Shah for their help during the early stages of our work on looped vision models.

Resources used in preparing this research were provided, in part, by the Province of Ontario, the Government of Canada through CIFAR, the Digital Research Alliance of Canada, and companies sponsoring the Vector Institute.

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

APPENDIX

# A TRAINING AND IMPLEMENTATION DETAILS

**Hardware and framework.** All models are trained on $4\times$H100 (80 GB) GPUs using the open-source NanoGPT training stack as a reference implementation.

**Data and tokens.** Unless otherwise specified, we run each experiment for 50,000 optimizer steps with a global batch size of 48 sequences and block size 1024 (context length). This corresponds to approximately $\sim$25B training tokens in total.

**Optimization.** We use AdamW with weight decay $2 \times 10^{-1}$, cosine learning-rate decay (per NanoGPT), peak learning rate $\mathtt{lr} = 6 \times 10^{-4}$, minimum learning rate $\mathtt{min\_lr} = 6 \times 10^{-5}$, and 4,000 warmup steps, which we found important for stability. Similar to observations in Geiping et al. (2025), we occasionally observe training instabilities at depth; warmup and cosine decay mitigate these in practice. Unless noted, other optimizer and training defaults follow NanoGPT.

**Model hyperparameters.** Following Saunshi et al. (2025), we use hidden size $d = 2048$ and $n_{\mathrm{heads}} = 32$ for all $(k\otimes L)$ configurations. The feed-forward dimension is $d_{\mathrm{ff}} = 5120$ with a standard two-layer GELU MLP. All normalizations are RMSNorm. We use learned positional embeddings added to token embeddings (NanoGPT default). We also tested RoPE Su et al. (2024) and observed slightly better small-scale performance, but we use learned positions for simplicity and efficiency. We tie input and output embeddings (weight tying).

**LoopFormer conditioning.** We use two embedding modules, one for normalized time $t \in [0, 1]$ and one for step size $\Delta t \in (0, 1]$. Each maps a scalar input to a $d$-dimensional conditioning vector via (i) fixed sinusoidal Fourier features of width $D_f = 256$ and max period 10,000, followed by (ii) a 2-layer MLP with hidden size $d$ and $\mathtt{SiLU}$ nonlinearity:

$$\phi(\tau) \;=\; \mathrm{MLP}\big([\cos(\tau\omega_1), \sin(\tau\omega_1), \dots, \cos(\tau\omega_{D_f/2}), \sin(\tau\omega_{D_f/2})]\big) \in \mathbb{R}^d,$$

where $\omega_k = \exp\big(-\frac{k-1}{D_f/2}\log 10{,}000\big)$ for $k = 1, \dots, D_f/2$. Given batchwise scalars $t$ and $\Delta t$, we compute $e_t = \phi(t)$ and $e_\Delta = \phi(\Delta t)$ and sum them to obtain the per-iteration conditioning signal $c = e_t + e_\Delta \in \mathbb{R}^d$.

Conditioning is applied inside each LoopFormer block via an AdaLN-style modulator: a small MLP takes $c$ and outputs $4d$ parameters, which we split into $(\alpha_{\mathrm{msa}}, \alpha_{\mathrm{mlp}}, \gamma_{\mathrm{msa}}, \gamma_{\mathrm{mlp}})$. We use RMSNorm (with no learned affinity) before MHSA and FFN, and apply multiplicative scaling and residual gating as

$$x \leftarrow x + \alpha_{\mathrm{msa}} \odot \mathrm{MHSA}\big(\mathrm{RMSNorm}(x) \odot (1 + \gamma_{\mathrm{msa}})\big),$$
$$x \leftarrow x + \alpha_{\mathrm{mlp}} \odot \mathrm{FFN}\big(\mathrm{RMSNorm}(x) \odot (1 + \gamma_{\mathrm{mlp}})\big),$$

broadcast over the sequence length. The modulator is a $\mathtt{SiLU}$ followed by a linear layer with output size $4d$, and is *zero-initialized* (weights and bias), ensuring the initial behavior matches the unmodulated backbone and that conditioning is learned stably.

**TMLT baseline.** For Time-Modulated Looped Transformers, we follow the authors' setup and condition each iteration on the loop index, implementing the timestep modulation as described in their paper.

# B ADDITIONAL EXPERIMENTS AND ABLATIONS

## B.1 COMPUTATIONAL OVERHEAD OF LOOPFORMER

Algorithm 1 uses a dual-trajectory objective: for every batch we compute (i) the full $L$-loop trajectory for the main LM loss, and (ii) a sampled shorter $M$-loop trajectory for shortcut-consistency. This design adds some overhead during training, however, during the inference, the computational

Table 2: Perplexity and zero-shot reasoning for $K=3$ models. We compare a 3-layer vanilla Transformer against LoopFormer evaluated at three inference budgets: $(3 \otimes 2)$, $(3 \otimes 4)$, and $(3 \otimes 8)$. LoopFormer matches vanilla performance at low budgets, and consistently improves with additional loops, substantially outperforming the vanilla baseline as compute increases, while retaining depth elasticity.

| | Params / FLOPs | Perplexity ↓ | | | Language Tasks (Acc) ↑ | | | | | | | | | | |
|---|---|---|---|---|---|---|---|---|---|---|---|---|---|---|---|
| | | Pile | FineWeb-Edu | OpenWebText | COPA | HS | LB | OBQA | PIQA | Race | SciQ | ARC | SIQA | WG | Avg Acc |
| Base $(3 \otimes 1)$ | 3x / 3x | 12.93 | 30.23 | 29.20 | 59 | 28.7 | 26.47 | 26.4 | 59.85 | 27.08 | 64.1 | 30.7 | 35.57 | 51.46 | 40.93 |
| LoopFormer - Ours $(3 \otimes 2)$ | 3x / 6x | 14.3 | 33.45 | 32.46 | 63 | 28.69 | 26.14 | 26.6 | 60.1 | 25.74 | 58.6 | 29.29 | 35.26 | 50.2 | 40.36 |
| LoopFormer - Ours $(3 \otimes 4)$ | 3x / 12x | 11.12 | 25.02 | 24.21 | 68 | 31 | 32.35 | 25.4 | 63.06 | 28.23 | 66.3 | 31.78 | 37.77 | 53.43 | 43.73 |
| LoopFormer - Ours $(3 \otimes 8)$ | 3x / 24x | 10.28 | 22.87 | 21.98 | 66 | 32.3 | 38.27 | 26.8 | 63.33 | 30.81 | 68 | 32.71 | 37.97 | 51.94 | 44.81 |

Flops and inference times of LoopFormer scale similar to that of a vanilla transformer. In the following we report the training overhead.

For clarity, we decompose the per-batch FLOPs into a loop-independent overhead plus a loop-dependent term. Let $C_{io}$ denote the FLOPs of the embedding / unembedding (non-loop compute), and let $C_1$ denote the FLOPs of one loop through the shared $K$-block stack. Then running $\ell$ loops costs

$$C(\ell) = C_{io} + \ell C_1.$$

LoopFormer computes a full $L$-loop trajectory plus one sampled shorter trajectory of length $M$, where $M \sim \text{Unif}\{1, \dots, L-1\}$. Thus the expected per-batch FLOPs are

$$C(L) + \mathbb{E}_M[C(M)] = (C_{io} + LC_1) + (C_{io} + \mathbb{E}[M] C_1).$$

Since $M$ is uniform on $\{1, \dots, L-1\}$, we have $\mathbb{E}[M] = L/2$. Plugging in,

$$C(L) + \mathbb{E}_M[C(M)] = 2C_{io} + (L + \tfrac{L}{2}) C_1 = 2C_{io} + \tfrac{3L}{2} C_1.$$

Relative to the fixed-loop/vanilla training cost $C(L) = C_{io} + LC_1$, the multiplicative overhead is

$$\frac{C(L) + \mathbb{E}_M[C(M)]}{C(L)} = \frac{2C_{io} + \tfrac{3L}{2} C_1}{C_{io} + LC_1},$$

which, for the regimes considered in this work, is approximately $1.5\times$ the FLOPs of fixed-loop training.

We confirm this empirically. Across our settings, LoopFormer requires about $1.5\times$ the training FLOPs of a fixed-loop or vanilla transformer baseline trained under the same token budget and number of iterations. In wall-clock time, this corresponds to an approximate $1.3\times$ slowdown in our setup (measured on 4×H100 GPUs with identical batch size, optimizer, and data). It is worth mentioning that other depth-elastic baselines studied in this work (e.g. TMLT-EE or Base-Loop-EE), also have the same overhead as our LoopFormer.

Overall, LoopFormer pays a modest training overhead to enable elastic-depth inference: a single shared-parameter model that remains robust under aggressive truncation and continues to refine representations as more loops are allocated. The inference on the other has a similar computational FLOPs to that of vanilla or fixed-loop baselines.

## B.2 COMPARISON WITH VANILLA TRANSFORMERS UNDER A SIMILAR PARAMETER BUDGET

Here we compare LoopFormer to a vanilla Transformer under a similar parameter budget. We fix the shared stack to $K = 3$ layers, and evaluate LoopFormer at different inference budgets, namely $(3 \otimes 2)$, $(3 \otimes 4)$, and $(3 \otimes 8)$. Table 2 reports perplexity and downstream language task accuracy for LoopFormer and a 3-layer vanilla Transformer with a similar number of parameters. We observe that at small compute (few loops), LoopFormer is on par with the vanilla baseline, while increasing the loop budget leads to consistent improvements, allowing our depth-elastic model to substantially outperform the vanilla Transformer.

Table 3: Perplexity and zero-shot reasoning under FLOPs-matched training and inference. We compare $(3 \otimes 8)$ looped models as well as a Base $(24 \otimes 1)$. To match total training FLOPs, LoopFormer is trained for 34k iterations (seeing ∼8B fewer Pile tokens), while baselines are trained for 50k iterations. Despite the reduced token budget, LoopFormer matches TMLT and remains competitive on reasoning across budgets, while retaining depth-elastic behavior.

| | Params / FLOPs | Perplexity ↓ | | | Language Tasks (Accuracy) ↑ | | | | | | | | | | |
|---|---|---|---|---|---|---|---|---|---|---|---|---|---|---|---|
| | | Pile | FineWeb-Edu | OpenWebText | COPA | HS | LB | OBQA | PIQA | Race | SciQ | ARC | SIQA | WG | Avg Acc |
| Base $(24 \otimes 1)$ | 24x / 24x | 9.49 | 20.7 | 20.08 | 61 | 35.04 | 41.96 | 27.6 | 66 | 29 | 70.1 | 33.43 | 38.18 | 50.4 | 45.27 |
| Base-Loop $(3 \otimes 8)$ | 3x / 24x | 10.91 | 24.53 | 24.53 | 61 | 30.46 | 34.68 | 27 | 63.22 | 28.8 | 63.7 | 31.71 | 38.43 | 49.8 | 42.88 |
| TMLT $(3 \otimes 8)$ | 3x / 24x | 10.38 | 22.87 | 21.99 | 65 | 32.34 | 39.06 | 27.8 | 63.11 | 29.67 | 69.8 | 31.67 | 36.95 | 51.54 | 44.69 |
| LoopFormer - Ours $(3 \otimes 8)$ | 3x / 24x | 10.71 | 23.66 | 22.78 | 64 | 31.96 | 37.11 | 25.6 | 62.8 | 29.91 | 68.5 | 31.68 | 37.93 | 52.57 | 44.21 |

## B.3 COMPARISON UNDER FLOPs-MATCHED TRAINING

As discussed in subsection B.1, elastic-depth training incurs approximately $1.5 \times$ the training FLOPs of vanilla or fixed-loop Transformers due to the additional sampled shortcut trajectory. To evaluate LoopFormer under a FLOPs-matched training regime, we train a LoopFormer model for 34k iterations (all other hyperparameters unchanged), resulting in roughly the same total training FLOPs as the baselines trained for 50k iterations. This compute-matched LoopFormer therefore sees about 8B fewer Pile tokens during training. Table 3 reports perplexity and zero-shot reasoning when both training and inference FLOPs are matched. Even under this reduced training budget, LoopFormer remains on par with TMLT and other baselines, while additionally providing depth-elastic inference.

## B.4 PYTORCH PSEUDOCODE FOR LOOPFORMER

Algorithm 3 presents a self-contained PyTorch-style pseudocode snippet for the core LoopFormer shared stack. The module consists of a `LoopFormerBlock`, which applies AdaLN-style modulation to the attention and MLP residual branches using the loop-conditioning vector $c$ (derived from $(t, \Delta t)$), and a `SharedBlock` that stacks $K$ such blocks to form the shared Transformer $\Phi_k$.

Within each loop iteration, `SharedBlock` applies all $K$ blocks sequentially to the token states $x$; this entire $K$-block stack is then reused across loops and repeated for $M$ or $L$ iterations during training or inference. This snippet clarifies how LoopFormer handles multiple blocks per loop and how the same shared stack supports variable compute budgets.

---

**Algorithm 3: PyTorch implementation of LoopFormer shared block**

---

```python
# Inputs:
#   x      : torch.Tensor [B, T, D] - token states.
#   c      : torch.Tensor [B, D]    - loop-conditioning vector.
#   config : model config with n_embd, etc.
#   depth  : int (K) - number of distinct Transformer blocks

class LoopFormerBlock(nn.Module):

    def __init__(self, config):
        super().__init__()
        d = config.n_embd

        self.ln_1 = nn.RMSNorm(d, elementwise_affine=False)
        self.attn = CausalSelfAttention(config)

        self.ln_2 = nn.RMSNorm(d, elementwise_affine=False)
        self.mlp  = MLP(config)

        # Conditioning -> (gate_msa, gate_mlp, scale_msa, scale_mlp)
        self.adaLN_modulation = nn.Sequential(
            nn.SiLU(),
            nn.Linear(d, 4 * d, bias=True),
        )

        # AdaLN-Zero init: start from identity updates
        nn.init.zeros_(self.adaLN_modulation[1].weight)
        nn.init.zeros_(self.adaLN_modulation[1].bias)

    def forward(self, x: torch.Tensor, c: torch.Tensor) -> torch.Tensor:
        gate_msa, gate_mlp, scale_msa, scale_mlp = \
            self.adaLN_modulation(c).chunk(4, dim=1)

        # Multi-head attention branch
        x = x + gate_msa.unsqueeze(1) * self.attn(
            self.ln_1(x) * (1.0 + scale_msa.unsqueeze(1))
        )

        # Feed-forward branch
        x = x + gate_mlp.unsqueeze(1) * self.mlp(
            self.ln_2(x) * (1.0 + scale_mlp.unsqueeze(1))
        )
        return x

class SharedBlock(nn.Module):

    def __init__(self, depth: int, config):
        super().__init__()
        self.blocks = nn.ModuleList(
            [LoopFormerBlock(config) for _ in range(depth)]
        )

    def forward(self, x: torch.Tensor, c: torch.Tensor) -> torch.Tensor:
        for blk in self.blocks:
            x = blk(x, c)
        return x
```

---