# OpenReview forum: "LoopFormer: Elastic-Depth Looped Transformers for Latent Reasoning via Shortcut Modulation"
_ICLR.cc/2026/Conference — ICLR 2026 Poster_

### Official Review · Reviewer_pdxW · 2025-10-16

**Soundness:** 4
**Presentation:** 3
**Contribution:** 3
**Rating:** 8
**Confidence:** 5

**Summary:**

This paper introduces LoopFormer, a looped Transformer architecture that can flexibly adapt its computational depth based on a user-specified budget, a feature termed "elastic depth". The model is designed to perform robustly across a range of loop iterations at inference time without needing to be retrained, addressing a key limitation of prior looped models.

Traditional looped Transformers are trained and evaluated with a fixed number of loop iterations. This rigidity means they cannot adapt to variable compute budgets; their internal representations tend to collapse or stagnate when evaluated at depths different from their training configuration, leading to degraded performance.

The paper introduces a shortcut-consistency training scheme that enables compute-budgeted inference (elastic depth) without retraining the model.
The work demonstrates that naive early-exiting in looped models leads to representational collapse, where hidden states stagnate across iterations. In contrast, LoopFormer's representations continue to evolve, showing that it uses additional depth effectively for refinement.

**Strengths:**

The paper introduces the novel concept of "elastic depth" for looped Transformers, creatively adapting ideas from diffusion models to frame iterative refinement as a continuous-time "thought trajectory" . This unique problem formulation and synthesis of ideas results in a highly original approach to budget-conditioned reasoning.

The work is of high quality, featuring a rigorous experimental setup with strong, appropriate baselines and a comprehensive evaluation on both perplexity and a wide range of reasoning tasks . The claims are further substantiated by an in-depth representational analysis that uses multiple metrics to convincingly demonstrate that LoopFormer avoids the representational collapse that plagues naive adaptive methods.

The paper's contribution is significant because it provides a practical solution to the important problem of adaptive computation, making efficient looped architectures more versatile and deployable.

**Weaknesses:**

The shortcut-consistency training algorithm requires two forward passes per batch (one for the full trajectory and one for a shortcut), which effectively doubles the training compute compared to baseline looped models . The paper acknowledges this "added training overhead" but does not analyze the trade-off. A key missing experiment is a comparison against a baseline model trained for twice as many steps, which would clarify if the inference-time flexibility is worth the significant increase in training cost.

The model's performance is highly sensitive to the choice of the inference-time step schedule, yet this schedule must be selected manually by the user1111. This introduces a required hyperparameter tuning step that complicates deployment and may prevent users from achieving the model's optimal performance for a given budget. The work would be stronger if it included a method to learn or predict an optimal schedule automatically.

The experiments are conducted on ~1B parameter models, and it is unclear if the training dynamics and performance benefits will scale to much larger, state-of-the-art foundation models.

**Questions:**

Your results show that performance is sensitive to the choice of the inference-time schedule. Could you provide a practical heuristic for selecting a high-performing schedule without an exhaustive search? Have you considered methods for learning an optimal, input-dependent schedule to automate this process? A response here would clarify if this is a minor tuning step or a significant practical hurdle.

The shortcut-consistency training algorithm appears to double the training compute by requiring two forward passes per batch . Could you provide an analysis of this trade-off? Specifically, how does LoopFormer compare against a baseline (e.g., TMLT) that is trained for twice as many steps using the same total compute budget? This would clarify whether the inference-time flexibility justifies the increased training cost.

---

> ### Author Response · Authors · 2025-11-23
>
> We thank the reviewer for their valuable comments, feedback, and insightful questions. We address them below. Moreover, We have also added a new Appendix B with additional ablations/experiments, and made minor clarifications in the main paper, highlighted in blue in the revised manuscript.
>
>
> **W1: “The shortcut-consistency training algorithm requires two forward passes per batch (one for the full trajectory and one for a shortcut), which effectively doubles the training compute compared to baseline looped models . The paper acknowledges this "added training overhead" but does not analyze the trade-off. A key missing experiment is a comparison against a baseline model trained for twice as many steps, which would clarify if the inference-time flexibility is worth the significant increase in training cost.”**
>
> We agree this practical point should be quantified. In each batch we always run the full L-loop trajectory, and we additionally sample one shorter M-loop trajectory for the shortcut-consistency term. Since only a single shorter trajectory is added per batch, the extra compute is about half of a full pass on average. Consequently, LoopFormer training costs roughly 1.5× the FLOPs of a fixed-loop/Base model, and about 1.3× the wall-clock time in our setup (measured on 4× H100s with identical batch size and token budget). Further details are added in Appendix B.1 of the revised file.
> To directly address the equal-compute comparison, we also trained a compute-matched baseline: a LoopFormer with K=3 for 34k iterations, which approximately matches the total training FLOPs of a TMLT trained for 50k iterations (therefore seeing ~8B fewer Pile tokens). This compute-matched LoopFormer reaches comparable performance to TMLT, while additionally providing depth elasticity, whereas TMLT remains fixed-depth. We include this comparison table in Appendix B.3; for convenience we reproduce it here:
>
>
> |  | Params / FLOPs | | **Perplexity ↓**   |  | |  |  |  | **Language Tasks (Acc) ↑** |  |  | |  |  |  |
> |---|---:|---:|---:|---:|---:|---:|---:|---:|---:|---:|---:|---:|---:|---:|---:|
> |  |  | Pile | FineWeb-Edu | OpenWebText | COPA | HS | LB | OBQA | PIQA | Race | SciQ | ARC | SIQA | WG | Avg Acc |
> | Base (24⊗1) | 24x / 24x | 9.49 | 20.7 | 20.08 | 61 | 35.04 | 41.96 | 27.6 | 66 | 29 | 70.1 | 33.43 | 38.18 | 50.4 | 45.27 |
> | Base-Loop (3⊗8) | 3x / 24x | 10.91 | 24.53 | 24.53 | 61 | 30.46 | 34.68 | 27 | 63.22 | 28.8 | 63.7 | 31.71 | 38.43 | 49.8 | 42.88 |
> | TMLT (3⊗8) | 3x / 24x | 10.38 | 22.87 | 21.99 | 65 | 32.34 | 39.06 | 27.8 | 63.11 | 29.67 | 69.8 | 31.67 | 36.95 | 51.54 | 44.69 |
> | LoopFormer – Ours (3⊗8) | 3x / 24x | 10.71 | 23.66 | 22.78 | 64 | 31.96 | 37.11 | 25.6 | 62.8 | 29.91 | 68.5 | 31.68 | 37.93 | 52.57 | 44.21 |
>
> **W2: “The model's performance is highly sensitive to the choice of the inference-time step schedule, yet this schedule must be selected manually by the user1111. This introduces a required hyperparameter tuning step that complicates deployment and may prevent users from achieving the model's optimal performance for a given budget. The work would be stronger if it included a method to learn or predict an optimal schedule automatically.”**
>
> We appreciate and agree with the reviewer’s feedback. Schedule sensitivity is an interesting axis, and developing an automatic, budget-aware (potentially input-dependent) scheduling policy to select Δ_M without manual tuning is a natural and promising future direction for this framework.
>
> **W3: “The experiments are conducted on ~1B parameter models, and it is unclear if the training dynamics and performance benefits will scale to much larger, state-of-the-art foundation models.”**
>
>
> We appreciate this point. Our experiments focus on ~1B-parameter models, and scaling LoopFormer to much larger foundation models trained on broader, more diverse token mixtures would be an exciting and valuable extension to further validate elasticity at scale.
>
> **Q1: “Your results show that performance is sensitive to the choice of the inference-time schedule. Could you provide a practical heuristic ... clarify if this is a minor tuning step or a significant practical hurdle.”**
>
> We agree that a learned, budget-aware scheduling policy would be a valuable extension. For example, a controller could select Δ_M adaptively based on the prompt, intermediate confidence/entropy, or how the response evolves across loops, so harder queries receive denser schedules while easier ones exit earlier. In this work, we focused on showing that simple fixed schedules already yield strong elasticity and consistent gains, and we have not yet explored learning such a policy. We view adaptive, input-dependent schedule learning as a promising extension and leave it for future work.
>
> **Q2: “The shortcut-consistency training algorithm appears to double... Specifically, how does LoopFormer compare ... justifies the increased training cost.”**
>
> Addressed in W1.

---

### Official Review · Reviewer_xXws · 2025-10-26

**Soundness:** 3
**Presentation:** 3
**Contribution:** 3
**Rating:** 8
**Confidence:** 3

**Summary:**

This paper presents LoopFormer, which allows reasoning under variable compute budgets. It extends prior looped Transformers by introducing time- and step-size modulation, where each iteration receives sinusoidal embeddings of layer index and step size to dynamically modulate RMSNorm and residual scaling.
A shortcut-consistency loss aligns representations across different loop lengths, enabling stable performance even with fewer inference steps.

**Strengths:**

* The paper is well written and easy to follow, with clear motivation and setups
* The motivation is clearly presented, and the transition from fixed-depth looped Transformers to elastic-depth design feels natural.
* Experiments are reasonably comprehensive, evaluating both variable loop lengths and the effect of the proposed shortcut-consistency loss.
* The main claims are well supported

**Weaknesses:**

* The degree of novelty is not bad but moderate. While the proposed elastic-depth formulation and shortcut-consistency loss are well designed, they extend existing time-modulated looped Transformer frameworks rather than introducing a fundamentally new paradigm.
* the paper does not provide theoretical intuition or analysis explaining why combining t and $\Delta t$ through sinusoidal modulation is a good choice here

**Questions:**

NA

---

> ### Author Response · Authors · 2025-11-23
>
> We thank the reviewer for their valuable comments and feedback. The main concerns are addressed below. Moreover, We have also added a new Appendix B with additional ablations/experiments, and made minor clarifications in the main paper, highlighted in blue in the revised manuscript.
>
> **W1: “The degree of novelty is not bad but moderate. While the proposed elastic-depth formulation and shortcut-consistency loss are well designed, they extend existing time-modulated looped Transformer frameworks rather than introducing a fundamentally new paradigm.”**
>
> We appreciate the reviewer’s perspective and agree our method builds on the looped/time-modulated Transformer line, but we believe the novelty is substantial in both capability and mechanism. First, LoopFormer introduces true depth elasticity for looped LMs, with stable performance across any inference budget M≤L. Prior looped or time-modulated models are trained for a fixed loop count and degrade sharply under truncation; LoopFormer is the first to make looped models reliably elastic, where fewer loops remain strong and additional loops continue refining representations. Second, this is enabled by two components used jointly: (1) conditioning each loop on both normalized time and step size, so shared blocks adapt their update magnitude to arbitrary schedules rather than treating all loops as equivalent; and (2) shortcut-consistency training, aligning short-trajectory representations with full-trajectory ones to prevent stagnation and loop-collapse. Finally, we provide targeted representation analyses (curvature, anisotropy, entropy, CKA over loops) that illustrate how elasticity emerges and why baselines fail. Overall, this combination of a new elastic-depth objective, schedule-aware architecture, and mechanistic evidence goes beyond an incremental variant.
>
> **W2: “the paper does not provide theoretical intuition or analysis explaining why combining t and Δt
> through sinusoidal modulation is a good choice here”**
>
> We acknowledge that we do not introduce a new theoretical proof that sinusoidal embeddings of (t,Δt) are optimal. Our design choice is deliberately aligned with standard practice in diffusion and continuous-time generative models, where sinusoidal / Fourier-feature time embeddings are widely used to condition networks on continuous trajectories and to generalize across different discretizations and step sizes (e.g., DDPM/DiT-style models and shortcut / one-step diffusion). To the best of our knowledge, this literature also does not claim unique optimality via formal proof; rather, sinusoidal embeddings are motivated as a robust, empirically effective parameterization that allows the network to represent smooth functions of time and maintain resolution-invariant behavior as step sizes change. We adopt the same mechanism and extend it to looped Transformers by conditioning on both normalized time t and step size Δt, since Δt is exactly what varies when compute budgets change. A deeper theoretical study of why sinusoidal embeddings are particularly effective for trajectory-conditioned looped models is an interesting direction for future work, but is orthogonal to our main contribution here: elastic-depth training with shortcut-consistency for looped Transformers.

---

> > ### Comment · Reviewer_xXws · 2025-11-28
> >
> > Thanks the author for responses, I will keep my score

---

### Official Review · Reviewer_exw9 · 2025-11-01

**Soundness:** 3
**Presentation:** 3
**Contribution:** 3
**Rating:** 8
**Confidence:** 3

**Summary:**

This paper tackles the limitation of looped Transformers, whose performance typically degrades when inference loops don't match the fixed number used in training. They propose LoopFormer which novelly condition each loop on both normalized time and step size to adapt to different computational depth. The model is trained on "shortcut-consistency" loss that forces shorter trajectories to match the representation of the full trajectory. They empirically show that the LoopFormer perform more robustly at smaller compute budgets than existing baselines.

**Strengths:**

1. The paper tackles a clear, practical, and important problem. Enabling flexible, elastic compute in parameter-efficient models like looped Transformers is a highly valuable research direction.
2. The paper is well written and easy to follow.
3. The experiments are thorough and convincing. In addition to strong task performance, the authors provide a compelling explanation for why LoopFormer works by analyzing metrics like curvature, anisotropy, and CKA similarity. They demonstrate that baselines suffer from representational stagnation (flat metrics, high CKA), while LoopFormer's representations continue to evolve and refine with each loop. This analysis significantly strengthens the paper beyond just reporting better numbers.

**Weaknesses:**

1. The training procedure (Algorithm 1) requires two forward passes per batch (for the full and short trajectories) to compute the consistency loss. This appears to roughly double the training cost compared to a standard looped model. The paper mentions this as a limitation but does not quantify it. A brief analysis of the training FLOPs/time overhead vs. a Base-Loop or TMLT baseline would be valuable for assessing the practical trade-offs.
2. The paper heavily emphasizes "latent reasoning," using terms like "thought trajectories." However, the benchmarks (COPA, PIQA, HellaSwag) are standard zero-shot LM evaluation tasks, not complex, multi-step algorithmic reasoning tasks. The model shows improved general performance (including perplexity), and the representation analysis shows refinement, not necessarily reasoning in a formal sense. So the terming could be a bit misleading to readers.

**Questions:**

1. What is the practical training overhead (e.g., in FLOPs or wall-clock time) of the dual forward pass required for the consistency loss, compared to a standard Base-Loop or TMLT baseline?
2. Given that Figure 5 shows significant performance variance based on the chosen step schedule Δ_M, have you explored methods for learning an optimal, budget-aware scheduling policy?
3. The model is trained to interpolate (M <= L). Have you tested its ability to extrapolate to M > L loops? Does performance continue to improve, or does it diverge given the t=1 training target?

---

> ### Author Response · Authors · 2025-11-23
>
> We thank the reviewer for their useful feedback and appreciation of our contributions. The main concerns are addressed below. We have also added a new Appendix B with additional ablations/experiments, and made minor clarifications in the main paper, highlighted in blue in the revised manuscript.
>
> **W1: “The training procedure (Algorithm 1) requires two forward passes per batch (for the full and short trajectories) to compute the consistency loss. This appears to roughly double the training cost compared to a standard looped model. The paper mentions this as a limitation but does not quantify it. A brief analysis of the training FLOPs/time overhead vs. a Base-Loop or TMLT baseline would be valuable for assessing the practical trade-offs.”**
>
> We agree this should be quantified. For each batch, we always run the full L-loop trajectory, and additionally sample one shorter M-loop trajectory to apply shortcut-consistency. Since only a single extra shorter trajectory is computed per batch, the added work is about half of a full pass on average. This results in approximately 1.5× the training FLOPs of a fixed-loop/Base model, and about a 1.3× wall-clock slowdown in our setup (measured on 4× H100 GPUs with identical batch size and token budget). We add these details in Appendix B.1 of the revised file.
>
> **W2: “The paper heavily emphasizes "latent reasoning," using terms like "thought trajectories." However, the benchmarks (COPA, PIQA, HellaSwag) are standard zero-shot LM evaluation tasks, not complex, multi-step algorithmic reasoning tasks. The model shows improved general performance (including perplexity), and the representation analysis shows refinement, not necessarily reasoning in a formal sense. So the terming could be a bit misleading to readers.”**
>
> We appreciate this nuance and agree the wording should be precise. Our use of “latent reasoning” / “thought trajectories” is intended to describe iterative representation refinement across loops, not to claim formal multi-step algorithmic reasoning. The key message is that, unlike fixed-loop baselines, LoopFormer continues to update and refine its internal state with more loops, which is why performance scales smoothly with compute. To avoid overstatement, we revised the text to clarify this interpretation and to tone down phrasing that could imply stronger reasoning claims, while keeping the focus on elastic-depth refinement.
>
> **Q1: “What is the practical training overhead (e.g., in FLOPs or wall-clock time) of the dual forward pass required for the consistency loss, compared to a standard Base-Loop or TMLT baseline?”**
>
> Answered in W1; the concrete FLOPs and wall-clock overhead are reported in Appendix B.1.
>
> **Q2: “Given that Figure 5 shows significant performance variance based on the chosen step schedule Δ_M, have you explored methods for learning an optimal, budget-aware scheduling policy?”**
>
> We agree there is room for a learned, budget-aware scheduling policy. For example, a controller could select Δ_M adaptively based on the prompt, intermediate confidence/entropy, or how the response evolves across loops, allocating denser schedules to harder queries and exiting earlier on easier ones. In this work we focused on showing that simple fixed schedules already provide strong elasticity and consistent gains, and we have not yet explored learning such a policy. We view adaptive, input-dependent scheduling as a promising extension and leave it for future work.
>
> **Q3: “The model is trained to interpolate (M <= L). Have you tested its ability to extrapolate to M > L loops? Does performance continue to improve, or does it diverge given the t=1 training target?”**
>
> Thanks for raising this interesting question.We did test extrapolation to M > L. In our current formulation, extrapolation can occur in two ways, and in both cases our limited experiments show divergence rather than continued improvement.
> Case 1: M > L but the total normalized time still sums to 1, i.e., we insert smaller unseen steps. For Δ = [1/8]*7 + [1/16]*2 (an unseen 1/16 step), perplexity rises from 10.28 (baseline Δ = [1/8]*8) to 22.07.
> Case 2: M > L and the total time exceeds 1, i.e., stepping past the training endpoint. Here divergence is stronger: Δ = [1/8]*9 yields perplexity 85.05, and Δ = [1/8]*8 + [1/32] yields 60.28.
> At its current stage, LoopFormer is designed for interpolation and does not support stable extrapolation beyond L loops; we agree this is an important direction for future work.

---

### Official Review · Reviewer_aD1e · 2025-11-02

**Soundness:** 2
**Presentation:** 3
**Contribution:** 2
**Rating:** 4
**Confidence:** 3

**Summary:**

This paper presents a new model composed of k blocks each trained to be repeated up to L times. The architecture adds in each block a side information about the scheduling to visit [0, 1] with M steps, and modulate a scale and gate layer before/after the multi-head attention and FFN blocks.

The training procedure uses a loss that combines the cross-entropy of the full-compute (L iterations), the cross-entropy of a lesser compute (M iterations), and a L2 between the pre-readout activations of both.

Experimental validation shows that this approach is not as good as a vanilla architecture for a given compute budget, but has a lesser parameter count.

**Strengths:**

The proposed architecture is reasonably simple and well motivated. The overall direction, transformer-like that can dynamically modulate compute, is important. The results showing the nice monotonicity of perplexity or accuracy vs. FLOPs (Fig 2) is great.

**Weaknesses:**

The experimental results are underwhelming, and maybe I missed the point, but it is unclear to me how this model improves wrt a vanilla transformer. The results are presented in such a way that the training budget is not equalized (if I am correct? I do not understand the first sentence of 4.1), the inference flops are, and the parameter counts are not, although this is generally not the limiting factor.

Training budgets should have been equalized (e.g. pick the nb of training iterations per model), and results should be summarized with scatter plots on e.g. an average accuracy vs. wall-clock time or flops frame.

The base model should be added as a point on Fig 2 (a) and (b)

**Questions:**

It is unclear in Fig 1 and in the loss definition of 3.3 how the method deals with multiple blocks. It looks as if there was only one there.

Regarding the training cost and the overall performance, can you clarify in what regime you see the usefulness of this model?

---

> ### Author Response · Authors · 2025-11-23
>
> We thank the reviewer for their useful feedback. The main concerns are addressed below. We have also added a new Appendix B with additional ablations/experiments, and made minor clarifications in the main paper, highlighted in blue in the revised manuscript.
>
> **C1: “The experimental results are underwhelming, and maybe I missed the point, but it is unclear to me how this model improves wrt a vanilla transformer.”**
>
> Our goal is not to outperform a non-shared (vanilla) Transformer at the same inference FLOPs; rather, it is to add elastic depth to looped Transformers, a setting where vanilla models are not the right comparator. Looped models are attractive because they provide large parameter savings via weight sharing, but existing looped LMs are fixed-depth: when evaluated at budgets different from training, their representations collapse and performance degrades. LoopFormer addresses this limitation by training across variable-length loop trajectories with time/step conditioning and shortcut-consistency, enabling a single shared-parameter model to run robustly at any user-specified budget M ≤ L without retraining.
>
> **C2: “The results are presented in such a way that the training budget is not equalized (if I am correct? I do not understand the first sentence of 4.1), the inference flops are, and the parameter counts are not, although this is generally not the limiting factor.”**
>
> We apologize for the ambiguity in the first sentence of Sec. 4.1 (“We train looped models under different parameters and compute budgets”). To clarify: we train LoopFormer and all looped baselines under different parameter budgets by varying K, i.e., the number of Transformer blocks in the shared stack Φ_k(·). This is the standard way to scale looped models and is the axis studied in Fig. 2 and Sec. 4.2.
> Regarding compute budgets: LoopFormer (and other depth-elastic looped variants) incur additional training cost because we optimize both full and shortcut trajectories. Concretely, this makes training about 1.5× higher in FLOPs and roughly 1.3× higher in wall-clock time than fixed-loop or vanilla Transformers. We have made this training-cost difference explicit in (Appendix B.1) to avoid confusion.
> (We address the suggestion on parameter matching in W1 below.)
>
> **W1: “Training budgets should have been equalized (e.g. pick the nb of training iterations per model), and results should be summarized with scatter plots on e.g. an average accuracy vs. wall-clock time or flops frame.”**
>
> We thank the reviewer for this suggestion. The added 1.5× training compute is the necessary cost of enabling depth elasticity; under this setting, LoopFormer and the other depth-elastic looped baselines are trained with matched training FLOPs and comparable wall-clock time. Moreover, all models in the paper are trained on the same set of tokens for 50,000 iterations: specifically, the exact same subset of The Pile totaling about 25B tokens. Figure 2 already provides FLOPs-equalized comparisons within the elastic-depth looped regime.
> To further address parameter matching, we additionally report a parameter-matched comparison where a 3-layer vanilla Transformer is compared to a 3-layer LoopFormer, evaluated under different inference budgets to show depth elasticity (Appendix B.2).  For convenience we reproduce the table here:
>
> |  | Params / FLOPs |  | **Perplexity ↓** |  |  |  |  |  | **Language Tasks (Acc) ↑** |  |  |  |  |  |  |
> |---|---:|---:|---:|---:|---:|---:|---:|---:|---:|---:|---:|---:|---:|---:|---:|
> |  |  | Pile | FineWeb-Edu | OpenWebText | COPA | HS | LB | OBQA | PIQA | Race | SciQ | ARC | SIQA | WG | Avg Acc |
> | Base (3⊗1) | 3x / 3x | 12.93 | 30.23 | 29.20 | 59 | 28.7 | 26.47 | 26.4 | 59.85 | 27.08 | 64.1 | 30.7 | 35.57 | 51.46 | 40.93 |
> | LoopFormer – Ours (3⊗2) | 3x / 6x | 14.3 | 33.45 | 32.46 | 63 | 28.69 | 26.14 | 26.6 | 60.1 | 25.74 | 58.6 | 29.29 | 35.26 | 50.2 | 40.36 |
> | LoopFormer – Ours (3⊗4) | 3x / 12x | 11.12 | 25.02 | 24.21 | 68 | 31 | 32.35 | 25.4 | 63.06 | 28.23 | 66.3 | 31.78 | 37.77 | 53.43 | 43.73 |
> | LoopFormer – Ours (3⊗8) | 3x / 24x | 10.28 | 22.87 | 21.98 | 66 | 32.3 | 38.27 | 26.8 | 63.33 | 30.81 | 68 | 32.71 | 37.97 | 51.94 | 44.81 |
>
>
> **W2: “The base model should be added as a point on Fig 2 (a) and (b)”**
>
> We appreciate this suggestion and agree it improves clarity. We have added the base model as points in Fig. 2(a,b). While doing so, we also found and fixed a plotting bug in Fig. 2(b): one benchmark (sciq) had been inadvertently omitted. After correcting this and adding the base-model point, the updated figure shows an even cleaner trend and further supports our claims. We have included the corrected plots in the revised submission.

---

> > ### Author Response · Authors · 2025-11-23
> >
> > **Q1: “It is unclear in Fig 1 and in the loss definition of 3.3 how the method deals with multiple blocks. It looks as if there was only one there.”**
> >
> > Thanks for pointing this out. We agree the figure and notation could be clearer. The model uses K distinct Transformer blocks inside the shared stack Φ_k, and each loop iteration applies the entire K-block stack. In Fig. 1 we inadvertently omitted a small “K×” marker next to the shared stack, which made it look like a single block; we have added this marker in the updated version.
> > Concretely, within each loop we iterate through the K blocks in order (as in prior looped / weight-shared Transformers), and this K-block application is repeated up to L times. Appendix B.4 provides the PyTorch implementation of LoopFormer.
> >
> > **Q2: “Regarding the training cost and the overall performance, can you clarify in what regime you see the usefulness of this model?”**
> >
> > LoopFormer is useful in regimes where parameter efficiency and flexible test-time compute both matter: when you want the memory/parameter savings of looped (weight-shared) Transformers, but also need to run the same model at different inference budgets M ≤ L without retraining or suffering collapse. In that setting, LoopFormer provides a single shared-parameter model that remains robust under aggressive loop truncation for fast inference, and keeps improving as more loops are allocated, enabling anytime prediction. The target use case is thus upgrading looped models into elastic, deployable systems across variable compute or latency constraints, rather than replacing vanilla Transformers at fixed compute.

---

### Meta-Review · Area_Chair_dXm9 · 2026-01-13

**Summary:**

The paper examines a critical limitation in looped Transformers: their inability to adapt to variable compute budgets at inference time. The proposed solution, LoopFormer, introduces a novel shortcut-consistency training scheme and a architecture that effectively allows the model to perform elastic inference where single looped model performs well across different budgets without retraining or degeneracy.

The reviews for the paper are generally positive. While the paper received one negative rating initially, the majority of reviewers (three ratings of 8) strongly supported the work. The negative review primarily concerned experimental fairness (equalizing training budgets) and baseline comparisons. In my opinion, the authors’ rebuttal addressed these specific concerns with additional baselines (Appendix B.2 and B.3), demonstrating that LoopFormer remains competitive with vanilla models while offering the unique advantage of depth elasticity. I recommend acceptance.

**Reviewer Concerns:**

Reviewer aD1e was the only review who gave negative score. The reviewer mentioned that the results were underwhelming and that training budgets were not normalized (LoopFormer uses more training compute). The authors directly provided the requested equalized comparisons (both parameter and compute-matched), which showed the method holds up.

**Reviewer Scores:**

Reviewer aD1e would probably increase their score given their primary concerns were addressed.

Rest of the reviewers would keep their relatively high scores since their concerns were addressed. Overall this paper will have scores on the higher side.

---

### Decision · Program_Chairs · 2026-01-26

Accept (Poster)